# Evaluation of the psychometric properties of the Episodic Disability Questionnaire (EDQ) among women living with HIV in the United Kingdom: A self-reported repeated measure study

Darren A. Brown OBE[1]*☯, Shema Tariq[2,3]☯, Marta Boffito[1], David Asboe[1], Ana Milinkovic[1], Nneka Nwokolo[1], Carol Flavell[4‡], Sophie Strachan[1,5‡], Lisa Avery[6,7]☯, Kelly K. O'Brien[8,9,10]☯, Richard Harding[11‡]

1 Chelsea and Westminster Hospital NHS Foundation Trust, London, United Kingdom, 2 University College London, London, United Kingdom, 3 Central and North West London NHS Foundation Trust, London, United Kingdom, 4 James Cook University, Townsville, Queensland, Australia, 5 Sophia Forum, London, United Kingdom, 6 University Health Network, Toronto, Canada, 7 Dalla Lana School of Public Health, University of Toronto, Toronto, Canada, 8 Department of Physical Therapy, Temerty Faculty of Medicine, University of Toronto, Toronto, Canada, 9 Institute of Health Policy, Management and Evaluation, Dalla Lana School of Public Health, University of Toronto, Toronto, Canada, 10 Rehabilitation Sciences Institute, University of Toronto, Toronto, Canada, 11 Cicely Saunders Institute, Florence Nightingale Faculty of Nursing, Midwifery and Palliative Care, King's College London, London, United Kingdom

☯ These authors contributed equally to this work.
‡ CF, SS and RH also contributed equally to this work
* darren.brown11@nhs.net

## Abstract

### Background

Disability is increasingly experienced by women ageing with HIV and multimorbidity. The Episodic Disability Questionnaire (EDQ) measures the presence, severity, and episodic nature of disability across six domains. We evaluated EDQ properties among women living with HIV in the United Kingdom.

### Methods

Participants in the Positive Transitions Through the Menopause (PRIME) study completed the EDQ at two timepoints (1 week apart), criterion measures (WHODAS 2.0, EQ-5D-5L, Work and Social Adjustment Scale), and a demographic questionnaire. We evaluated internal consistency, test-retest reliability, measurement precision (Minimum Detectable Change (MDC) 95%), and construct validity. We assessed disability prevalence using WHODAS 2.0 (moderate threshold) and Equality Act Disability Definition (severe threshold).

**Data availability statement:** All data are available from: https://doi.org/10.5061/dryad.tht76hfdd.

**Funding:** This study is funded by a British HIV Association (BHIVA) research award (BHIVA/5020/2020/Brown) who played no role in study design. Kelly K. O'Brien is supported by a Canada Research Chair in Episodic Disability and Rehabilitation (CRC-2022-00510).

**Competing interests:** The authors have declared that no competing interests exist.

## Results

Of 104 participants (median age 56 years, 65% Black ethnicity), 93 (89%) completed the EDQ twice. Median duration since HIV diagnosis was 23 years; 98% had undetectable viral loads and 86% reported multimorbidity. Cronbach's alpha ranged from 0.83 (social domain) to 0.92 (daily activities domain). ICC ranged from 0.70 (physical domain) to 0.91 (daily activities domain). Precision varied, highest in daily activities (MDC95%: 6.10) and lowest in mental-emotional domains (MDC95%: 11.52). The EDQ met 80% (n = 47/59) of construct validity hypotheses. Disability prevalence was 79.81% (95%CI 70.57, 86.79) moderate and 41.75% (32.24, 51.88) severe.

## Conclusions

The EDQ possesses internal consistency, test-retest reliability, and construct validity with varied precision among women living with HIV. Disability prevalence in this sample was higher than in the general population. The EDQ offers value for research, clinical practice, and national policy by enabling measurement and description of disability, supporting intervention evaluation, and informing priority-setting and healthcare service planning for women living with HIV in the UK.

## Introduction

With access to effective antiretroviral therapy, people living with HIV are increasingly reaching older age in the United Kingdom (UK) and are more likely to experience multimorbidity and disability [1–3], presenting significant health-related challenges. Disability is defined by adults living with HIV as any physical, cognitive, mental-emotional symptoms, difficulties with day-to-day activities, challenges to social inclusion, and uncertainty about future health, that may persist or fluctuate on a daily basis or over the longer course of living with HIV [4,5].

Functioning and disability are important for monitoring the performance of health strategies in health systems [6]. Measuring disability is additionally important for determining the impact of health challenges, improving communication between providers and patients, and evaluating interventions [7–9].

The Episodic Disability Questionnaire (EDQ) is a generic 35-item patient-reported outcome measure (PROM) of disability [10] informed by the episodic disability framework [4,11]. Episodic disability is conceptualised as any health-related challenges that may fluctuate or persist over time when living with a health condition [4,11]. The EDQ is derived from the HIV-specific Short-Form HIV Disability Questionnaire (SF-HDQ) [12] and HIV Disability Questionnaire (HDQ) [13], which both possess validity, reliability, and sensibility for use among adults living with HIV in Canada, Ireland, the United States (US) and the UK [14–18]. The EDQ has demonstrated internal consistency, construct validity, and test–retest reliability, with limited precision when administered electronically across a sample comprised of primarily men (83%) living with HIV [10].

In the UK, people with disabilities experience inequities in social inclusion, education, employment, accessing health services, living standards, housing, well-being,

loneliness, and are more commonly victims of crime than the general population [19,20]. National policy recognises that women living with HIV experience distinct health challenges and inequities [21]. Women are more likely to experience disability than men [19,22,23] and women living with HIV in the UK experience higher disability severity compared to men [3]. Considering the gendered dimensions of disability, the under-representation of women in HIV research [24], and policy requirements for improved evidence, monitoring and tailored support for women living with HIV [21], it is essential to evaluate the psychometric properties of disability measurement tools among women. We aimed to assess the EDQ for its ability to measure disability experienced by women living with HIV in the UK, specifically internal consistency, test-retest reliability, precision of measurement, and construct validity. We also aimed to measure disability prevalence and report disability profiles among women living with HIV.

## Methods

We conducted a cross-sectional repeated measurement study involving the administration of the EDQ and criterion measures with women living with HIV in England, UK. We followed established COnsensus-based Standards for the selection of health Measurement Instruments (COSMIN) guidelines, a set of consensus-based reporting recommendations for primary studies of the measurement properties of patient-reported outcome measures, to evaluate and report the measurement properties of the EDQ [25].

### Study setting

This study was conducted at ten NHS outpatient HIV clinical settings in three cities in England, UK. We received ethics approval from the London City & East Research Ethics Committee (REC reference: 22/PR/1483) and Health Research Authority (IRAS: 318781). This study was included in the NIHR CRN Portfolio (CPMS ID: 54554).

### Participants

We recruited women living with HIV who had previously participated in the Positive Transitions Through the Menopause (PRIME) study, regardless of menopausal status [26]. Potential participants were eligible if they had participated in PRIME, and were therefore living with HIV, female sex, aged 45–60 years on entry to PRIME, and had provided consent to be contacted about future research. Exclusion criteria were the inability to give consent or complete questionnaires in English. Participants were recruited via their local clinical care team. Written informed consent was obtained from participants at the initial study information and consent page of the questionnaire administration.

### Data collection

Between March 2023 to January 2024 we electronically administered the EDQ followed by three criterion measures (World Health Organization Disability Assessment Schedule (WHODAS 2.0) [27], EQ-5D-5L [28], and Work and Social Adjustment Scale (WSAS) [29]) and a demographic questionnaire using Qualtrics software (https://www.qualtrics.com). Participants completed the questionnaires in-person via a computer or tablet at the recruiting clinical site, or remotely via a link in an email or Short Message Service (SMS) text. One week later, we emailed participants with a link requesting them to complete the EDQ only. At this time, we asked whether participants had a major change in their health status since their last EDQ completion and if yes, to describe this change. Study authors did not have access to information that could identify individual participants during or after data collection.

### Questionnaires

**Episodic Disability Questionnaire.** The EDQ is a 35-item generic PROM of disability encompassing six domains: i) physical (10 items); ii) cognitive (3 items); iii) mental-emotional (5 items), iv) uncertainty about future health (5 items), v)

difficulties carrying out day-to-day activities (5 items), and vi) challenges to social inclusion (7 items) [10]. For each item, individuals report the extent they experience a specific health-related challenge on the day of assessment (*severity scale* of 0–4) and whether each challenge fluctuated over the past week (*episodic score* yes or no) [10]. The presence scale is calculated by dichotomising the severity scale as either present (severity 1–4) or absent (severity of 0) [10]. The EDQ was derived from the preceding SF-HDQ [12]. Severity and presence domain scores are calculated using the algorithm derived from the previous Rasch analysis (score range: 0–100) [10,12]. Higher scores indicate greater presence, severity and episodic nature of disability [10].

**World Health Organization disability assessment schedule.** The WHODAS 2.0 is a 12-item generic PROM of functioning and disability, directly linked to the International Classification of Functioning, Disability and Health (ICF) [27,30]. The WHODAS 2.0 assesses the level of difficulty individuals experience performing specific functions over the past 30 days, encompassing six domains: i) cognition, ii) mobility, iii) self-care, iv) getting along, v) life activities and vi) participation. Individuals answer a 5-point ordinal scale (range 0–4) with higher scores indicating greater difficulty completing the activity [27]. In simple scoring, scores are summed to provide a value out of 48, with higher scores suggesting greater disability [27]. Complex scoring (or item response theory-based scoring) weights individual item severity, providing a disability range from 0 (no disability) to 100 (total disability) [27]. The WHODAS possesses internal consistency, test–retest reliability, validity and cross-cultural applicability [30,31]. The WHODAS is validated in people with chronic health conditions [32] and people living with HIV [33]. Categorisation thresholds have been developed to identify people living with HIV experiencing disability (score ≥2) [3,34,35], and any level of functional limitation (score ≥1) [36,37]. In the UK, scores ≥2 have been defined as "moderate" disability among people living with HIV [3].

**EQ-5D-5L.** The EuroQOL five dimensions five-level questionnaire is a generic PROM of health-related quality of life, encompassing five domains: i) mobility; ii) self-care; iii) usual activities; iv) pain/discomfort; and v) anxiety/depression [28,38]. Each domain has 5 levels: no problems, slight problems, moderate problems, severe problems, and extreme problems. The digits for the five dimensions can be combined into a 5-digit number that describes the patient's health state, or represented by a single summary number known as the index value, with a range approximately −0.285 to 1.000 [28,39]. The EQ-5D-5L is an extensively used generic measure of health-related quality of life in HIV research and across different diseases worldwide [40,41]. Psychometric data support its use [42], and it has been used within national HIV reporting and the PRIME study [26,43].

**Work and Social Adjustment Scale.** The WSAS is a generic PROM of perceived impairment in work or social functioning resulting from a health problem [29]. The 5 domains are scored on an ordinal scale from 0 (not at all) to 8 (very severe), to identify challenges in: i) work; ii) home management; iii) social leisure activities; iv) private leisure activities; and v) relationships with others. Scores range from 0 to 40, with lower scores indicating better adjustment. WSAS scores above 20 suggest severe functional impairment, scores between 10 and 20 suggest moderately severe and scores below 10 suggest mild [29,44]. The WSAS has demonstrated high internal consistency among people living with HIV [44] and is used to assess functional impairment among people living with HIV and painful peripheral neuropathy [45].

**Demographic questionnaire.** The demographic questionnaire included 35 items encompassing demographic (e.g., age, sex, gender, ethnicity), health (e.g., time since HIV diagnosis, viral load, concurrent health conditions, menopause, and general health status), and social characteristics (e.g., living arrangements, work status, social security or benefits), and the UK Equality Act Disability Definition (EADD) questions [46,47]. The EADD reflects legal definitions in the Equality Act 2010 and defines a person as disabled if they have a physical or mental impairment, and that impairment has a substantial and long-term adverse effect on their ability to carry out day-to-day activities [48]. The classification questions making up the EADD are: (a) "*do you have any physical or mental health conditions or illnesses lasting or expecting to last 12-months or more?*"; (b) "*do any of your conditions of illnesses reduce your ability to carry out day-to-day activities?*". A person is counted as disabled if they answer "*yes*" to both classification questions. The EADD has been used to define "severe" disability among people living with HIV in the UK [3].

 

## Analysis

We calculated median (interquartile ranges (IQR)) EDQ scores aligned to previous analysis [10]. Severity and presence domain scores (range: 0–100) were calculated using the algorithm established from the previous Rasch analysis [10,12]. Episodic domain scores (range: 0–100) included a simple sum transformation [10]. Higher scores indicated greater presence, severity and episodic nature of disability [10]. We calculated median WHODAS 2.0 domain scores, EQ-5D-5L scores and WSAS scores as per guidelines. For the demographic questionnaire, we calculated descriptive statistics including frequencies (%) for categorical variables and median and IQR for continuous variables. Analyses were conducted in R version 4.4.0 using the *psych* package.

**Internal consistency.** We calculated Cronbach's alpha (severity domain) and Kuder-Richardson-20 (KR-20) (presence and episodic domains) for time 1 (T1) with 95% confidence intervals (CI) (≥0.7 acceptable) [10,49].

**Test-retest reliability.** We calculated Intra Class Correlations (ICCs) with 95% CI using T1 and time 2 (T2) EDQ scores based on Shrout and Fleiss' ICC2 (absolute agreement with random raters) (lower bound CI of > 0.7 acceptable) [10,49]. We calculated ICCs for the sample that indicated no change in health status between T1 and T2 [10]. Our test–retest assessment included only the EDQ presence and severity scales, consistent with previous analysis [10], as the episodic scale assesses for disability fluctuations over the past week and consistency in this scale was not expected.

**Measurement precision.** We calculated the Standardised Error of Measurement (SEM) using Wywrich criteria [50], $SEM = SD\sqrt{(1 - \rho_{ICC})}$ where $\rho_{ICC}$ is the test-retest reliability. The Minimum Detectable Change (MDC) was calculated for 95% CI (MDC95%) using the method proposed by Haley [51], $MDC_{1-\alpha} = z_{1-\alpha/2}\hat{\sigma}_{baseline}\sqrt{2(1 - \rho)}$ where ρ is the test-retest reliability ICC, 1 – α is the level of confidence, and $\hat{\sigma}_{baseline}$ is the standard deviation of the measure at T1 [10].

**Construct validity.** A priori hypotheses were formulated based on previous EDQ validation studies and clinical understanding [10], predicting specific correlational patterns between EDQ domains and the criterion measures (WHODAS 2.0, EQ-5D-5L, WSAS), as well as known group differences. We examined correlations for 59 total *a priori* hypotheses: 6 primary and 53 secondary. Primary hypotheses theorised relationships between EDQ domains' severity scores and WHODAS 2.0 total scores. Secondary hypothesis theorised relationships between EDQ and WHODAS, EQ-5D-5L, and WSAS criterion measure sub-scales, and known groups of participants completing EDQ "on a good day" and with ≥2 concurrent health conditions [10]. The full list of *a priori* hypotheses is available in S1 Fig. We calculated Spearman correlation coefficients to test associations between continuous EDQ scores and criterion measures, and Wilcoxon rank sum tests to compare EDQ scores across ordinal criterion measures. Spearman correlation coefficients of |≥ 0.30|, |≥ 0.50| and |≥ 0.70| were defined as 'weak', 'moderate' and 'strong', respectively [10,52]. If the lower bound of the confidence interval for the Spearman correlation was greater than the pre-specified level, the criteria were considered to be met. Wilcoxon rank sum test for a test of difference in medians between the known groups was significant at a level of <0.05, for the criteria to be considered met. Construct validity was defined as ≥75% confirmed hypotheses [10,52].

**Disability prevalence.** We estimated disability prevalence frequencies (%) and 95% CI, representing a range of severity [53]. The threshold for experiencing "moderate" disability was defined as WHODAS scores ≥2, representing at least two mild/moderate or one moderate/severe limitation on WHODAS items [3,34–37]. The threshold for experiencing "severe" disability was defined as self-rating "yes" to both EADD classification questions [3].

**Disability profile.** We calculated the frequency (%) for WHODAS simple scores in response to all 12 items, presence of any functional limitation (score ≥1) per WHODAS domain, WHODAS total number of limitations, and WHODAS difficulty levels [3]. For WHODAS simple and complex sum scores, we calculated mean (standard deviation (SD)) and median (IQR as 25th-75th percentile) to align with normative data and existing literature [3,54,55]. We calculated median, lower quartile (LQ) and upper quartile (UQ), and range for EDQ presence, severity, and episodic scores per EDQ domain [3,10].

**Sample size.** A sample of 85 participants was required to detect a weak correlation |r = 0.30| between the EDQ and criterion scores, with 0.80 power and alpha of 0.05 [10,56]. To account for incomplete questionnaire responses, attrition at T2, and recruiting 25% eligible participants from each site, our targeted sample size was 104 women living with HIV.

## Results

One hundred and four participants completed the questionnaires at T1, of which 93 (89%) completed the EDQ at T2, a median of 11 days (IQR 8–19) after T1. Of the 93 completing T2, 59 (63%) participants reported no change in health status and were included in the test-retest reliability assessment. Among the 59 included, 51 (86%) reported completing T1 and T2 questionnaires on a good day, and 8 (14%) reported a bad day for both T1 and T2.

### Participant characteristics

Participants' demographic, health and social characteristics are provided in Table 1 and S1 Table. All participants were female sex assigned at birth, median age 56 years (IQR 54–58), with a median duration of 23 years since HIV diagnosis (IQR 18–27). Most participants were of Black, Black British, Caribbean or African ethnicity n = 67 (65%), taking anti-retroviral therapy n = 102 (99%), had an undetectable viral load <200 copies/ml n = 88 (98%), and were living with a median of 5 comorbidities (IQR 3–8), representing multi-morbidity (≥2 co-morbidities) n = 89 (86%).

### Criterion measures

WHODAS "simple" sum scores were mean 12.6 out of 48 (SD; 11.9), median 9.0 (IQR; 2.0–22.2). WHODAS "complex" sum scores were mean 26.3 out of 100 (SD; 24.8), median 18.8 (IQR; 4.2–46.3). The EQ-5D-5L median index scores were 0.7 (IQR: 0.5–0.8). See S2 Table for frequencies (%) for each ED-5D-5L domain/item. The WSAS scores were mean 9.9 out of 40 (SD;11.8), median 5 (IQR: 0–16).

### Internal consistency

The EDQ met criteria for internal consistency across domains in the presence, severity and episodic scales (ICC ≥0.7). Cronbach's alpha for EDQ severity scores ranged from 0.83 (social domain) to 0.92 (daily activities domain), for EDQ presence scores ranged from 0.75 (uncertainty domain) to 0.90 (daily activities domain), and for EDQ episodic scores ranged from 0.70 (social domain) to 0.81 (daily activities domain) (Table 2). Lower bound CIs were ≥0.7, except for the uncertainty domain (0.67) of the EDQ presence scale and social domain (0.61) of the EDQ episodic scale (Table 2).

### Test-retest reliability

Overall, the EDQ met criteria for test-retest reliability for EDQ severity domains with ICCs ranging from 0.70 (physical domain) to 0.91 (daily activities domain) and for EDQ presence domains ranging from 0.72 (physical domain) to 0.89 (daily activities domain) (Table 3). Lower bound CIs were >0.7, except for the physical domain (0.53) of the EDQ severity scale and physical (0.57), cognitive (0.60) and uncertainty (0.63) domains of the EDQ presence scale (Table 3).

### Precision

The EDQ severity scale for each domain demonstrated the highest precision, whereby the MDC95% ranged from 6.10 (daily activities domain) to 11.52 (mental-emotional domain), followed by the presence scale, whereby MDC95% ranged from 12.84 (social inclusion domain) to 21.32 (cognitive domain). The episodic scale MDC95% ranged from 13.75 (social inclusion domain) to 27.58 (cognitive domain) (Table 4).

### Construct validity

Eighty percent (47/59) of all hypotheses were met, including all six primary hypotheses (100%) and 41/53 (77%) secondary hypotheses confirmed, shown in S1 Fig, supporting construct validity for use with women living with HIV in the UK.

**Table 1. Participants' demographic, health, and social characteristics.**

| Age (N = 104) | Median (25th, 75th percentile) |
|---|---|
| Median age (years) | 56 (54, 58) |
| **Sex** (n = 103) | **Number (%)** |
| Female sex assigned at birth | 103 (100%) |
| **Gender** (n = 103) | **Number (%)** |
| Gender Identity same as sex assigned at birth | 103 (100%) |
| **Ethnicity** (n = 103) | **Number (%)** |
| Black, Black British, Caribbean or African – African | 63 (61%) |
| Black, Black British, Caribbean or African – Caribbean | 3 (3%) |
| Black, Black British, Caribbean or African – Any other Black, Black British, Caribbean or African background | 1 (1%) |
| White – Any other White background | 12 (12%) |
| White – English, Welsh, Scottish, Northern Irish or British | 11 (11%) |
| White – Irish | 2 (2%) |
| Mixed or multiple ethnic groups – White and Black African | 3 (3%) |
| Mixed or multiple ethnic groups – Any other Mixed or Multiple backgrounds | 2 (2%) |
| Asian or Asian British – Any other Asian background | 3 (3%) |
| Asian or Asian British – Indian | 1 (1%) |
| Other ethnic group – Hispanic or Latino/a/x | 1 (1%) |
| Other ethnic group – Any other ethnic group | 1 (1%) |
| **Sexual orientation** (n = 103) | **Number (%)** |
| Straight or heterosexual | 102 (99%) |
| Bisexual | 1 (1%) |
| **Number of comorbidities** (N = 104) | **Median (25th, 75th percentile)** |
| Median number of comorbidities | 5 (3, 8) |
| **Multi-morbidity** (N = 104) | **Number (%)** |
| ≥2 co-morbidities | 89 (86%) |
| **Duration living with HIV** (n = 101) | **Median (25th, 75th percentile)** |
| Median years | 23 (18, 27) |
| **Most recent CD4 count** (n = 103) | **Number (%)** |
| <200 cells/mm³ | 6 (6%) |
| 201 - 349 cells/ mm³ | 6 (6%) |
| 350 - 499 cells/ mm³ | 11 (11%) |
| >500 cells/ mm³ | 46 (45%) |
| Don't know | 34 (33%) |
| **Most recent viral load** (n = 90) | **Number (%)** |
| <50 copies/ml – also known as Undetectable | 80 (89%) |
| 50 to 200 copies/ml – also known as Undetectable | 8 (9%) |
| 200 to 1000 copies/ml | 1 (1%) |
| >1000 copies/ml | 1 (1%) |
| **Taking antiretroviral therapy** (n = 103) | **Number (%)** |
| Yes | 102 (99%) |
| **Menopause status** [a] (n = 101) | **Number (%)** |
| Pre-menopausal | 2 (2%) |
| Peri-menopausal | 16 (16%) |
| Post-menopausal | 83 (82%) |
| **In receipt of benefits or social security supports** (n = 104) | **Number (%)** |

*(Continued)*

**Table 1.** (Continued)

| Age (N = 104) | Median (25th, 75th percentile) |
|---|---|
| Yes | 40 (38%) |
| **Economic activity** (N = 104) | **Number (%)** |
| Economically active | 76 (73%) |
| Economically inactive | 28 (27%) |
| **Educational attainment** (n = 103) | **Number (%)** |
| University degree level or above | 56 (54%) |
| O Levels, GCSEs or equivalent qualifications at age 16 | 20 (19%) |
| A Levels or equivalent education at age 18 | 19 (18%) |
| Finished education with no qualifications | 8 (8%) |
| **Has children** (n = 103) | **Number (%)** |
| Yes | 79 (77%) |
| **Lives alone or alone with pets** (n = 103) | **Number (%)** |
| Yes | 35 (34%) |

[a] Menopause status was categorised based on self-reported menstrual pattern.

**Table 2. Internal consistency for Episodic Disability Questionnaire (EDQ) domain scores (N = 104 participants).**

| EDQ Domain | EDQ Severity Cronbach's alpha (95% CI) | EDQ Presence KR-20 (95% CI) | EDQ Episodic KR-20 (95% CI) |
|---|---|---|---|
| **Physical** | 0.87 (0.83, 0.91) | 0.80 (0.74, 0.85) | 0.78 (0.71, 0.84) |
| **Cognitive** | 0.89 (0.85, 0.92) | 0.81 (0.74, 0.87) | 0.79 (0.71, 0.85) |
| **Mental-emotional** | 0.91 (0.88, 0.93) | 0.82 (0.76, 0.87) | 0.80 (0.73, 0.85) |
| **Uncertainty** | 0.86 (0.81, 0.90) | 0.75 (0.67, 0.82) | 0.80 (0.73, 0.85) |
| **Daily activities** | 0.92 (0.89, 0.94) | 0.90 (0.86, 0.93) | 0.81 (0.74, 0.86) |
| **Social inclusion** | 0.83 (0.77, 0.87) | 0.79 (0.73, 0.85) | 0.70 (0.61, 0.78) |

**Table 3. Test-retest reliability for the Episodic Disability Questionnaire (EDQ) domains for severity and presence scales (n = 59).**

| EDQ Domain | EDQ Severity ICC (95% CI) | EDQ Presence ICC (95% CI) | EDQ Episodic* ICC (95% CI) |
|---|---|---|---|
| **Physical** | 0.70 (0.53, 0.81) | 0.72 (0.57, 0.82) | 0.69 (0.47, 0.82) |
| **Cognitive** | 0.81 (0.70, 0.88) | 0.74 (0.60, 0.84) | 0.36 (0.12, 0.56) |
| **Mental-emotional** | 0.78 (0.65, 0.86) | 0.84 (0.75, 0.90) | 0.49 (0.27, 0.66) |
| **Uncertainty** | 0.89 (0.82, 0.94) | 0.76 (0.63, 0.85) | 0.31 (0.07, 0.52) |
| **Daily activities** | 0.91 (0.86, 0.95) | 0.89 (0.82, 0.93) | 0.20 (−0.05, 0.43) |
| **Social inclusion** | 0.87 (0.78, 0.92) | 0.82 (0.72, 0.89) | 0.25 (−0.01, 0.47) |

* For information only as consistency is not expected in the episodic scale **Measurement.**

## Disability prevalence

The estimated prevalence of moderate disability was 79.81% (95%CI: 70.57, 86.79) with n = 83 scoring ≥2 on WHODAS. The estimated prevalence of severe disability was 41.75% (95%CI: 32.24, 51.88) with n = 43 self-rating "yes" to both EADD classification questions.

**Table 4. Minimum Detectable Change (MDC) for Episodic Disability Questionnaire (EDQ) scales (N = 104 participants).**

| EDQ Domain | EDQ Severity SEM (MDC95%) | EDQ Presence SEM (MDC95%) | EDQ Episodic SEM (MDC95%) |
|---|---|---|---|
| Physical | 9.31 (18.24) | 13.25 (25.98) | 14.00 (27.45) |
| Cognitive | 10.38 (20.34) | 21.32 (41.79) | 27.58 (54.06) |
| Mental-emotional | 11.52 (22.59) | 14.27 (27.98) | 21.93 (42.97) |
| Uncertainty | 6.92 (13.56) | 14.55 (28.52) | 22.55 (44.21) |
| Daily activities | 6.10 (11.96) | 13.71 (26.87) | 23.24 (45.56) |
| Social inclusion | 6.75 (13.24) | 12.84 (25.16) | 13.75 (26.95) |

## Disability profile

The WHODAS simple scores in response to all 12 items are shown in S3 Table. Frequency of any functional limitation (score ≥1) within each of the six WHODAS disability domains were: challenges to social participation (n = 129, 62%), mobility challenges (n = 122, 59%), challenges to life activities (n = 116, 56%), cognitive health challenges (n = 103, 50%), challenges getting along (n = 90, 43%), and challenges with self-care (n = 72, 35%). Any level of functional limitation (WHODAS score ≥1) was reported by n = 90 (87%), whereby n = 9 (9%) scored one limitation, n = 10 (10%) two limitations, n = 7 (7%) three limitations, n = 64 (62%) four or more limitations, and n = 19 (18%) scoring all twelve limitations. Difficulty levels across all WHODAS items were "no difficulty' n=616 (49%), "mild difficulty" n = 203 (16%), "moderate difficulty" n = 247 (20%), "severe difficulty" n = 114 (9%), and "extreme difficulty/cannot do" n = 68 (5%). The T1 EDQ domain scores are shown in Table 5. The most severe, present, and episodic EDQ disability domains at T1 were "uncertainty", "uncertainty", and "physical symptoms and impairments" respectively, with n = 84 (81%) reported completing the EDQ on a good day at T1.

## Discussion

The EDQ exhibits internal consistency, test-retest reliability, and construct validity with varied precision among women living with HIV in the UK. This study extends previous evaluations of the EDQ's measurement properties, which were conducted largely with samples composed of men living with HIV [10,12,14–18]. These results are the first known to focus on the EDQ measurement properties with women.

The EDQ presence, severity and episodic scales demonstrated internal consistency with all lower bound CIs of Cronbach's alphas ≥0.70, apart from the uncertainty domain in the EDQ presence scale (0.67) and social domain in the EDQ

**Table 5. Median time 1 EDQ domain scores (N = 104 participants).**

| EDQ Domain | EDQ Severity Score Median (LQ, UQ) | EDQ Presence Score Median (LQ, UQ) | EDQ Episodic Score Median (LQ, UQ) |
|---|---|---|---|
| Physical | 38.0 (23.0, 46.2) | 59.0 (37.0, 68.2) | 20.0 (0.0, 50.0) [a] |
| Cognitive | 20.0 (0.0, 43.5) | 67.0 (0.0, 100.0) | 0.0 (0.0, 33.0) |
| Mental-emotional | 37.0 (18.0, 53.0) | 77.0 (22.0, 100.0) | 0.0 (0.0, 40.0) |
| Uncertainty | 42.0 (30.0, 55.5) [a] | 78.0 (59.0, 100.0) [a] | 0.0 (0.0, 20.0) |
| Daily activities | 21.0 (0.0, 38.5) | 42.0 (0.0, 100.0) | 0.0 (0.0, 20.0) |
| Social inclusion | 27.0 (8.0, 41.5) | 44.0 (18.0, 67.0) | 0.0 (0.0, 14.0) |

Higher scores indicate greater severity, presence and episodic nature of disability.

[a] indicates the highest scores across domains.

episodic scale (0.61). These findings suggest that the items operate cohesively and together reflect the broader construct of disability among women living with HIV in the UK. Our findings are consistent with earlier EDQ internal consistency analysis in Canada, Ireland, the UK and the US, whereby Cronbach's alphas ranged from 0.72 to 0.91 [10], while extending this evidence base to women living with HIV in the UK, a population not previously evaluated.

Our analysis of test-retest reliability demonstrated that the EDQ severity and presence scales achieved ICCs > 0.70, reflecting temporal stability. Lower bound CIs for all ICCs were >0.70, except the physical (0.53) domain of the EDQ severity scale and physical (0.57), cognitive (0.60) and uncertainty (0.63) domains of the EDQ presence scale. We assessed test–retest reliability for the EDQ presence and severity scales only, as the episodic scale aims to capture fluctuations within the past week and was not expected to remain stable. Results confirm this expectation, with lower ICCs for the EDQ episodic scale, reflecting the episodic nature of disability. These lower ICCs represent not a failure of reliability within the episodic scale but instead its sensitivity to change. This evaluation relied on only two measurement occasions; consequently, the day-to-day variability of disability may influence how T2 EDQ scores are interpreted and could lead to conservative estimates of temporal stability. Incorporating additional repeated assessments in future longitudinal work would strengthen understanding of how the EDQ performs across routine HIV care visits. The EDQ has been proposed as a useful tool for group-level monitoring and programme evaluation [10]. Ultimately, clinicians must determine what degree of measurement error is acceptable in clinical practice, depending on how EDQ results are intended to inform care decisions, service referrals, or eligibility for disability benefits [10,14,57]).

A lower MDC95% indicates greater precision, meaning smaller changes in scores can be confidently considered true changes rather than measurement error. The EDQ is most precise in detecting changes in the daily activities domain and least precise in the uncertainty domain. Among women living with HIV, the EDQ severity scale demonstrated higher precision, underscoring its potential utility for interpreting meaningful changes in disability over time. The severity scale offers clinicians and researchers a reference point for discerning minimal yet significant changes in disability longitudinally, surpassing day-to-day variability and measurement error.

Construct validity was supported, with 80% of all *a priori* hypotheses confirmed, including all primary and 77% of secondary hypotheses. This level of confirmation, alongside a conservative approach to evaluating both primary and secondary criteria, reinforces the EDQ's capacity to measure disability consistent with previous analysis [10,16]. Considering gendered dimensions of disability, these findings provide clinicians and researchers assurance of the EDQ's ability to measure disability experienced by women living with HIV in the UK.

Our disability prevalence estimates among this sample of working-age women living with HIV in the UK demonstrate that 4 in 5 (79.81%) women experienced moderate disability, and approximately 2 in 5 (41.75%) experienced severe disability. Identifying individuals with less severe disability is important, as focused public health interventions towards those with mild to moderate disability may generate additional gains in population health [58]. The severe disability threshold (EADD) follows the core definition of disability in the Equality Act 2010, permitting comparison to the general population. Our disability estimates exceed proportions of disability among women and working-age adults in the UK general population, trending towards proportions among state-pension age adults [19,59]. They also exceed disability prevalence among a sample comprised primarily of men living with HIV in the UK [3]. Notably, 1 in 6 (18%) participants reported via the EADD that their conditions or illnesses limited their ability to carry out day-to-day activities, despite indicating no physical or mental health conditions expected to last 12-months or more. Given that all participants were living with HIV, this may represent underreporting of severe disability prevalence and highlights potential limitations of duration-based criteria in capturing episodic or fluctuating disability experiences. This sample of women also lived with more concurrent health conditions (median 5) compared to previous estimates among men (median 2) [3], underscoring the gendered complexity of ageing with HIV and multimorbidity. The high prevalence of disability in this sample underscores the value of PROMs capable of accounting for episodic health challenges, particularly among women ageing with HIV.

Our findings demonstrate that women living with HIV experience multidimensional and episodic disability in the context of antiretroviral therapy and viral suppression. This study also provides the first known disability profiles of women living with HIV in the UK, capturing the breadth of disability by quantifying the magnitude or severity of impact, beyond a dichotomised reporting of those who are disabled and those who are not [58]. Relative to WHODAS population norms [27,54], disability scores in this sample were elevated. Mean WHODAS simple scores were approximately three to four times higher than general population averages, while mean WHODAS complex scores were four to six times higher [27,54]. Median scores were similarly elevated, reflecting the severity of disability experienced by women living with HIV in the UK.

Our results show that the most severe and present disability domain was uncertainty about future health. This aligns with previous research identifying uncertainty as the domain with the highest severity and presence scores [10,15–18], and indicates the importance of measuring uncertainty as a core component of disability experiences [60]. Consistent with UK data, physical symptoms and impairments were the most episodic and second most severe domain, experienced by women in this sample [10,16], emphasising the importance of screening, measuring and addressing these health challenges. Chronic pain of joints (45%) and soft tissue (42%) were the most frequent concurrent health conditions (S1 Table), with mobility impairments, social participation challenges and limitations with life activities being the most common WHODAS activity limitations (S3 Table). Musculoskeletal pain is common during perimenopause [61], highlighting the importance of detailed clinical history and awareness of the broad impacts of menopause transition. Furthermore, this sample of women reported higher rates of ≥1 WHODAS limitations than samples comprising mostly men, and higher rates of four or more activity limitations [3]. Physical symptoms and impairments can adversely affect daily functioning and reduce health-related quality of life [62]. Despite HIV care cascade successes in the UK [63], addressing the broader health, functioning, and well-being needs of women living with HIV requires disability-inclusive systems, interventions and multi-disciplinary care teams capable of providing comprehensive rehabilitation.

## Implications for practice and research

The EDQ can be used clinically to describe disability experienced by women living with HIV, determine the impact of health challenges, improve communication between providers and patients, and evaluate the effect of interventions [10]. Clinicians and women living with HIV can use the EDQ to better understand which disability dimensions pose challenges and better direct personalised interventions and rehabilitation. The EDQ is also unique in its ability to measure and describe the episodic nature of health challenges experienced over time [10,64]. Our findings indicate that women living with HIV in the UK are experiencing and self-reporting high levels of disability and multimorbidity. This warrants the inclusion of disability as a core outcome within national HIV surveillance, such as the UK Health Security Agency (UKHSA) annual reporting [1] and the Positive Voices survey [43], using validated tools to estimate prevalence and distribution of disability. Given that disability is associated with poorer health outcomes, fewer economic opportunities, and increased risk of poverty [65], the availability of valid and reliable disability data is essential for informing national strategies and action plans [21] and addressing negative societal attitudes [66]. Such data are critical for identifying priority areas and policy targets, guiding service planning and resource allocation, evaluating whether a population's health needs are met, and assessing effective intervention coverage. Moreover, disability data are required for monitoring progress against the United Nations (UN) Sustainable Development Goals (SDGs) and for fulfilling obligations under the Convention on the Rights of Persons with Disabilities (CRPD) [67–70].

## Strengths and limitations

We recruited ethnically diverse women living and ageing with HIV across the UK who had participated in the PRIME study, whereby the majority of participants were peri- or post-menopausal. Menopausal experiences of women in the UK are multifaceted, spanning physical, emotional, and social dimensions of health [71], and our results bring attention to the disability experiences of women living with HIV during the menopause. Our results may not be generalisable to all women living with HIV or transferable to low or middle-income countries. Future research should explore the EDQ's performance

with broader populations of women living with HIV, and our results can support sample size estimations. This assessment of EDQ properties involved electronic administration, and results may not be transferable to other modes of administration, such as paper-based administration [10]. Future research should explore the EDQ's performance when administered via different or mixed modalities, and investigate potential mode effects, as accessibility for all populations may require mixed-mode administration. Women represent over a third of people receiving HIV care in the UK in 2024, with 48% aged 50 years and over [1]. Consequently, these data are representative of a large and growing population of women living with HIV in the UK, addressing chronic underrepresentation of women, racially minoritised people and older people in HIV research [24,72]. Future longitudinal studies using the EDQ are necessary to further confirm its responsiveness to change over longer periods and to establish minimal clinically important difference (MCID) values.

## Conclusions

The EDQ demonstrated internal consistency, test-retest reliability, and construct validity with varied precision among women living with HIV. Disability prevalence and severity in this sample were higher than in the general population. The EDQ offers value for research, clinical practice, and national policy by enabling measurement and description of disability, supporting intervention evaluation, and informing priority-setting and healthcare service planning for women living with HIV in the UK.

## Supporting information

**S1 Table. Participants' additional demographic, health, and social characteristics.**
(DOCX)

**S2 Table. EQ-5D-5L frequency (%) of scores per domain.**
(DOCX)

**S3 Table. WHODAS 2.0 12-item frequency (%) of participants scoring per difficulty level.**
(DOCX)

**S1 Fig. Construct validity hypotheses testing and results of the Episodic Disability Questionnaire (EDQ).**
(PDF)

## Acknowledgments

We express our deepest gratitude to all the women living with HIV who participated in this research, sharing their experiences and providing valuable insights into various aspects of their health and functioning. We also thank the research team, including co-investigators and collaborators from ten NHS outpatient HIV clinical settings recruiting participants, and the Sophia Forum for their dedicated efforts in providing patient and public involvement, for this evaluation of disability among PRIME study participants, the UK's largest sample of women living with HIV.

## Author contributions

**Conceptualization:** Darren A. Brown OBE, Shema Tariq.

**Formal analysis:** Darren A. Brown OBE, Lisa Avery.

**Funding acquisition:** Darren A. Brown OBE, Shema Tariq, Kelly K. O'Brien, Richard Harding.

**Investigation:** Darren A. Brown OBE.

**Methodology:** Darren A. Brown OBE, Shema Tariq, Marta Boffito, David Asboe, Ana Milinkovic, Nneka Nwokolo, Carol Flavell, Sophie Strachan, Lisa Avery, Kelly K. O'Brien, Richard Harding.

**Project administration:** Darren A. Brown OBE.

**Supervision:** Kelly K. O'Brien, Richard Harding.

**Writing – original draft:** Darren A. Brown OBE, Shema Tariq, Marta Boffito, David Asboe, Ana Milinkovic, Nneka Nwokolo, Carol Flavell, Sophie Strachan, Lisa Avery, Kelly K. O'Brien, Richard Harding.

**Writing – review & editing:** Darren A. Brown OBE, Shema Tariq, Marta Boffito, David Asboe, Ana Milinkovic, Nneka Nwokolo, Carol Flavell, Sophie Strachan, Lisa Avery, Kelly K. O'Brien, Richard Harding.

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
