## [Decision Letter · Decision Letter 0]

20 Jan 2026

PONE-D-25-57930Evaluation of the psychometric properties of the Episodic Disability Questionnaire (EDQ) among women living with HIV in the United Kingdom: a self-reported repeated measure studyPLOS One

Dear Dr. Brown,

Thank you for submitting your manuscript to PLOS ONE. After careful consideration, we feel that it has merit but does not fully meet PLOS ONE’s publication criteria as it currently stands. Therefore, we invite you to submit a revised version of the manuscript that addresses the points raised during the review process.

If applicable, we recommend that you deposit your laboratory protocols in protocols.io to enhance the reproducibility of your results. Protocols.io assigns your protocol its own identifier (DOI) so that it can be cited independently in the future. For instructions see: https://journals.plos.org/plosone/s/submission-guidelines#loc-laboratory-protocols. Additionally, PLOS ONE offers an option for publishing peer-reviewed Lab Protocol articles, which describe protocols hosted on protocols.io. Read more information on sharing protocols at . Additionally, PLOS ONE offers an option for publishing peer-reviewed Lab Protocol articles, which describe protocols hosted on protocols.io. Read more information on sharing protocols at https://plos.org/protocols?utm_medium=editorial-email&utm_source=authorletters&utm_campaign=protocols..

We look forward to receiving your revised manuscript.

Kind regards,

Mariagrazia Benassi

Academic Editor

PLOS One

Journal Requirements:

- https://doi.org/10.1186/s12879-023-08958-7

In your revision ensure you cite all your sources (including your own works), and quote or rephrase any duplicated text outside the methods section. Further consideration is dependent on these concerns being addressed.

5. We note that your Data Availability Statement is currently as follows: All relevant data are within the manuscript and in Supporting Information files.

Additional Editor Comments:

Both the reviewers and I have carefully read your study.As you can see from the reviewers' comments, we considered your study highly relevant, clear and well-written but important revisions are needed to fully meet PLOS ONE's pubblication criteria.In particular, the reviewers expressed concerns and requested clarification regarding the methods section and the interpretation of the results.More details are needed to describe the methodology adopted in each analysis and the Rasch model procedures used. These aspects are also important because they strengthen the interpretation of the results.Please respond appropriately to each point raised by the two reviewers, indicating any changes you made to the text.

Reviewers' comments:

Reviewer's Responses to Questions

**Comments to the Author**

1. Is the manuscript technically sound, and do the data support the conclusions?

Reviewer #1: Yes

Reviewer #2: Yes

2. Has the statistical analysis been performed appropriately and rigorously? 

Reviewer #1: Yes

Reviewer #2: Yes

3. Have the authors made all data underlying the findings in their manuscript fully available?

Reviewer #1: Yes

Reviewer #2: Yes

4. Is the manuscript presented in an intelligible fashion and written in standard English?

Reviewer #1: No

Reviewer #2: Yes

5. Review Comments to the Author

Reviewer #1: Overall Assessment: The manuscript addresses a critical area in HIV care and women's health. The specific focus on women living with HIV, and the use of the EDQ to capture episodic disability, makes it distinctive and impactful. The adherence to COSMIN guidelines and the comprehensive psychometric evaluation are definite strengths.

Pros:

1. High Clinical and Social Relevance: The study addresses disability in women living with HIV, a population often underrepresented in research, particularly concerning episodic disability and multimorbidity. The focus on aging with HIV further enhances its relevance.

2. Strong Methodological Rigor:

o COSMIN Adherence: Explicitly stating adherence to COSMIN guidelines indicates a high standard for psychometric evaluation.

o Repeated Measures Design: The two-timepoint data collection for test-retest reliability is crucial for stability assessment.

o Use of Criterion Measures: Employing well-established measures like WHODAS 2.0, EQ-5D-5L, and WSAS for construct validity strengthens the findings.

3. Specific Population Focus: Targeting women living with HIV in the UK allows for nuanced insights into their specific experiences of disability, contrasting with previous studies predominantly on men.

Areas for Improvement (with specific suggestions):

1. Abstract Conciseness and Impact:

- Issue: The abstract, particularly the Methods and Results sections, is quite dense and could be streamlined to enhance clarity and impact for high-impact journals where abstracts are often read first. Phrases like "Cronbach alpha ≥0.7" or "ICC >0.7" are vague when specific ranges are available.

- Suggestion:

- Methods: Instead of "Cronbach alpha ≥0.7," state the observed range (e.g., "Cronbach alpha ranged from 0.83 to 0.92"). Similarly, use the observed ICC range.

- Results: Quantify "varied precision" (e.g., "Precision varied, highest in daily activities (MDC95%: 6.10) and lowest in mental-emotional (MDC95%: 11.52)").

- Typos: Correct the obvious typo "Cronbach alpha 30.7" to "≥0.7" (or the actual range) in the abstract.

- Overall: Aim for more active voice and direct phrasing to convey key findings efficiently.

2. Clarity on Construct Validity (Methods Section):

- Issue: The methods section states, "We examined correlations for 59 total a priori hypotheses: 6 primary and 53 secondary." While the outcome ("80%... hypotheses met") is in the abstract and results, the process of how these hypotheses were generated and tested is not fully detailed in the Methods.

- Suggestion: Briefly describe the nature of these a priori hypotheses in the Methods section. For example, "A priori hypotheses were formulated based on previous EDQ validation studies and clinical understanding, predicting specific correlational patterns between EDQ domains and the criterion measures (WHODAS 2.0, EQ-5D-5L, WSAS), as well as known group differences (e.g., by concurrent health conditions)." Referencing the Supplementary File for the full list is good, but a summary of how they were tested (e.g., using Spearman correlations and Wilcoxon rank sum tests, as later mentioned implicitly) should be here.

3. Measurement Precision Interpretation:

- Issue: The section on MDC95% provides the definition and values but could benefit from a clearer interpretation of what these values mean for the EDQ in practical terms. What does an MDC95% of 6.10 versus 11.52 imply for detecting meaningful change in clinical practice or research?

- Suggestion: In the "Measurement Precision" section of the Methods, or in the Discussion when elaborating on precision results, add a sentence or two explaining the practical implications of these MDC values. For example, "A lower MDC95% indicates greater precision, meaning smaller changes in scores can be confidently considered true changes rather than measurement error. Thus, the EDQ is most precise in detecting changes in [domain with lowest MDC] and least precise in [domain with highest MDC]."

4. Discussion of Episodic Scale Test-Retest:

- Issue: The methods state, "Our test-retest assessment focused on EDQ presence and severity scales, as the episodic scale refers to fluctuations in disability in the past week, and we did not expect consistency in this scale." However, Table 3 does present ICCs for the EDQ Episodic scale, some of which are quite low (e.g., Daily activities: 0.20).

- Suggestion: Reconcile this. Either remove the episodic scale ICCs from Table 3 if consistency was genuinely not expected and not relevant, or, more likely, explicitly discuss these lower ICCs in the Results and Discussion. If they are low because the scale is designed to capture fluctuations, then explain that this reflects the nature of episodic disability and is not necessarily a "failure" of reliability for that specific scale component, but rather an expected outcome demonstrating its sensitivity to change. This would strengthen the understanding of episodic disability measurement.

5. Deeper Discussion of Limitations and Future Research:

- Issue: While limitations are acknowledged, the discussion could expand on how these limitations affect interpretation and what specific future research is needed. For example, the limitation about electronic administration not being generalizable to paper-based could be explored further.

- Suggestion: For each key limitation (e.g., self-report, electronic administration, sample size for broader populations), suggest concrete future studies or methodological approaches. For instance: "Future research should explore the EDQ's performance when administered via paper-and-pencil, as accessibility for all populations may require mixed-mode administration, and investigate potential mode effects." Or "Longitudinal studies using the EDQ are necessary to further confirm its responsiveness to change over longer periods and to establish minimal clinically important difference (MCID) values."

6. Minor Presentation Points:

- "Table 1: Participants' demographic, health, and social characteristics": This table is very extensive. While good for transparency, consider if some less critical categories can be moved to supplementary material to keep the main text focused, especially if space is a concern for a high-impact journal.

By addressing these points, the manuscript will be greatly enhanced its readability for publication.

Reviewer #2: Review of the article PONE-D-25-57930 “Evaluation of the psychometric properties of the Episodic Disability Questionnaire (EDQ) among women living with HIV in the United Kingdom: a self-reported repeatedmeasure study”

The article entitled “Evaluation of the psychometric properties of the Episodic Disability Questionnaire (EDQ) among women living with HIV in the United Kingdom: a self-reported repeated measure study” evaluates the psychometric properties of the EDQ among women living with HIV in the United Kingdom using both classical analyses and analyses within the framework of Rasch models. On the whole, the article is clear and well-written, and it fulfils the aims and scopes of PLOS ONE. However, in my opinion some modifications are needed for the article to be published.

The main point that deserves attention is the part concerning the Rasch analyses. By reading the section “Results”, it seems to me that traditional analyses were run on each of the six domains of the EDQ, separately for severity, presence, and episodic nature of disability. Rasch analyses were also run on each of the six dimensions, separately for severity and presence, whereas Rasch analyses were not run for episodic nature of disability. This understanding, which I do not know if it is correct or not, resulted from reading the section “Results”, but not from the previous sections “Questionnaires” and “Analysis”. Moreover, several aspects concerning the Rasch analyses are missing and/or should be better clarified:

- To which Rasch analysis algorithms do the authors refer on line 195? I think the authors used the Rasch model for dichotomous items for the analysis of presence and episodic score, and a Rasch model for polytomous items for the analysis of severity but, with respect to the latter, it is not clear if they used the rating scale model (Andrich, 1978) or the partial credit model (Masters, 1982). The authors should make this clear.

- The authors should explain to which scores they refer when writing “score range: 0-100” in lines 195-196. From this paragraph, my guess is that the authors refer to linear transformations of the ability/health measures of individuals obtained by applying the Rasch model but, from line 257, it seems that the raw score are transformed to range from 0 to 100.

- Were the assumptions of the Rasch models verified? Specifically, did the authors verify that the six dimensions of the EDQ, in both the analysis on severity and the analysis of presence, were unidimensional? If they did not, the authors could use principal component analysis of the residuals to investigate unidimensionality (Smith, 2002). Did the authors verify the assumption of local independence?

- A Rasch analysis of polytomous items, conducted either with the rating scale model or with the partial credit model, allows for investigating if the rating scale functions as intended, that is, if higher response categories actually reflect higher levels of the latent variable. To this aim, it is needed that the thresholds (i.e., the points on the latent variable at which two adjacent categories are equally likely) are ordered increasingly (Andrich, 2010, 2011; Colledani et al., 2025). The authors should state if, in the Rasch analysis on polytomous items, the thresholds were ordered increasingly.

- The authors should add and interpret Rasch-based statistics of the functioning of the items such as outfit and/or infit (Linacre, 2002).

- The authors investigated internal consistency using Cronbach’s α (in the analysis of polytomous items) and KR-20 (in the analysis of dichotomous items). Since the authors also estimated Rasch models, they could report the Rasch-based measure of internal consistency called person-separation reliability, which applies to both dichotomous and polytomous items and has been found to be a better estimate of internal consistency than Cronbach’s α and KR-20 (Anselmi et al., 2019).

References

Andrich, D. (1978). A rating scale formulation for ordered response categories. Psychometrika, 43(4), 561–573. https://doi.org/10.1007/BF02293814

Andrich, D. (2010). Understanding the response structure and process in the polytomous Rasch model. In M. L. Nering & R. Ostini (Eds.), Handbook of polytomous item response theory models (pp. 123-152). Routledge. https://doi.org/10.4324/9780203861264

Andrich, D. (2011). Rating scales and Rasch measurement. Expert Review of Pharmacoeconomics & Outcomes Research, 11(5), 571–585. https://doi.org/10.1586/erp.11.59

Anselmi, P., Colledani, D., & Robusto, E. (2019). A Comparison of classical and modern measures of internal consistency. Frontiers in Psychology, 10, 2714. https://doi.org/10.3389/fpsyg.2019.02714

Colledani, D., González Pizzio, A. P., Devita, M., & Anselmi, P. (2025). Investigating the Functioning of Rating Scales with Rasch Models. Assessment, 32(3), 434–446. https://doi.org/10.1177/10731911241245792

Linacre, J. M. (2002). What do Infit and Outfit, Mean-square and Standardized mean? Rasch Measurement Transactions, 16(2), 878.

Masters, G. N. (1982). A Rasch model for partial credit scoring. Psychometrika, 47,149–174. http://dx.doi.org/10.1007/BF02296272

Smith, E. V., Jr., (2002). Detecting and evaluating the impact of multidimensionality using item fit statistics and principal component analysis of residuals. Journal of Applied Measurement, 3(2), 205–231.

Lines 287-288. It is written “We derived a priori hypotheses from earlier construct validity assessments of the EDQ [10].” The authors should briefly provide a description of their a priori hypotheses and how they were derived from earlier construct validity assessments of the EDQ

Minor points

Line 130: The authors could provide a brief definition of “episodic disability”.

Lines 152-155: The authors should provide a brief description of the guidelines they refer to.

Lines 235-236: It is written “The WSAS is a unidimensional scale, suggesting the scores cannot be compared across groups of different health conditions”. To my understanding, it is stated that the unidimensional nature of the WSAS suggests that the scores cannot be compared across groups of different health conditions. Is my understanding correct? If so, the authors should clarify why the unidimensional nature of the WSAS suggests that the scores cannot be compared across groups of different health conditions.

Line 262. If the authors used R packages, they should make explicit which packages they used.

Line 386: “were;” should be replaced with “were:”.

In the Section “Strengths and Limitations”, the authors should make more explicit which of the discussed aspects are strengths and which are limitations.

6. PLOS authors have the option to publish the peer review history of their article (what does this mean?). If published, this will include your full peer review and any attached files.). If published, this will include your full peer review and any attached files.

.

Reviewer #1: No

Reviewer #2: **Yes:** Pasquale AnselmiPasquale Anselmi

---

## [Author Response · Author response to Decision Letter 1]

25 Feb 2026

We would like to sincerely thank the reviewers for their thoughtful, constructive, and engaged feedback on our manuscript. We greatly appreciate the time and expertise invested in reviewing our work. The comments were insightful and have strengthened the clarity, rigour, and overall contribution of the paper.

---

## [Decision Letter · Decision Letter 1]

13 Apr 2026

Evaluation of the psychometric properties of the Episodic Disability Questionnaire (EDQ) among women living with HIV in the United Kingdom: a self-reported repeated measure study

PONE-D-25-57930R1

Dear Dr. Brown,

We’re pleased to inform you that your manuscript has been judged scientifically suitable for publication and will be formally accepted for publication once it meets all outstanding technical requirements.

An invoice will be generated when your article is formally accepted. Please note, if your institution has a publishing partnership with PLOS and your article meets the relevant criteria, all or part of your publication costs will be covered. Please make sure your user information is up-to-date by logging into Editorial Manager at Editorial Manager® and clicking the ‘Update My Information' link at the top of the page. For questions related to billing, please contact  and clicking the ‘Update My Information' link at the top of the page. For questions related to billing, please contact billing support..

Kind regards,

Mariagrazia Benassi

Academic Editor

PLOS One

Additional Editor Comments (optional):

I believe the revisions you submitted are adequate in response to both reviewers' comments. Thank you for your effective response, which allowed you to improve the work by taking the reviewers' suggestions into account.

Reviewers' comments:

Reviewer's Responses to Questions

**Comments to the Author**

1. If the authors have adequately addressed your comments raised in a previous round of review and you feel that this manuscript is now acceptable for publication, you may indicate that here to bypass the “Comments to the Author” section, enter your conflict of interest statement in the “Confidential to Editor” section, and submit your "Accept" recommendation.

Reviewer #2: All comments have been addressed

2. Is the manuscript technically sound, and do the data support the conclusions?

Reviewer #2: Yes

3. Has the statistical analysis been performed appropriately and rigorously? 

Reviewer #2: Yes

4. Have the authors made all data underlying the findings in their manuscript fully available?

Reviewer #2: Yes

5. Is the manuscript presented in an intelligible fashion and written in standard English?

Reviewer #2: Yes

6. Review Comments to the Author

Reviewer #2: I would like to thank the authors for the care they took in reviewing the manuscript. To my opinion, the manuscript in the current form is appropriate for pubblication in PLoS One.

7. PLOS authors have the option to publish the peer review history of their article (what does this mean?). If published, this will include your full peer review and any attached files.). If published, this will include your full peer review and any attached files.

.

Reviewer #2: **Yes:** Pasquale AnselmiPasquale Anselmi

---

## [Editor Report · Acceptance letter]

PONE-D-25-57930R1

PLOS One

Dear Dr. Brown OBE,

I'm pleased to inform you that your manuscript has been deemed suitable for publication in PLOS One. Congratulations! Your manuscript is now being handed over to our production team.

Kind regards,

on behalf of

Dr. Mariagrazia Benassi

Academic Editor

PLOS One